# Symmetry-Breaking Bifurcations of the Information Bottleneck and Related Problems

**DOI:** 10.3390/e24091231

**Published:** 2022-09-02

**Authors:** Albert E. Parker, Alexander G. Dimitrov

**Affiliations:** 1Center for Biofilm Engineering, Department of Mathematical Sciences, Montana State University, Bozeman, MT 59717, USA; 2Department of Mathematics and Statistics, Washington State University Vancouver, Vancouver, WA 98686, USA

**Keywords:** information bottleneck, optimization, annealing, gradient flow, bifurcations, symmetry

## Abstract

In this paper, we investigate the bifurcations of solutions to a class of degenerate constrained optimization problems. This study was motivated by the Information Bottleneck and Information Distortion problems, which have been used to successfully cluster data in many different applications. In the problems we discuss in this paper, the distortion function is not a linear function of the quantizer. This leads to a challenging annealing optimization problem, which we recast as a fixed-point dynamics problem of a gradient flow of a related dynamical system. The gradient system possesses an $S_{N}$ symmetry due to its invariance in relabeling representative classes. Its flow hence passes through a series of bifurcations with specific symmetry breaks. Here, we show that the dynamical system related to the Information Bottleneck problem has an additional spurious symmetry that requires more-challenging analysis of the symmetry-breaking bifurcation. For the Information Bottleneck, we determine that when bifurcations occur, they are only of pitchfork type, and we give conditions that determine the stability of the bifurcating branches. We relate the existence of subcritical bifurcations to the existence of first-order phase transitions in the corresponding distortion function as a function of the annealing parameter, and provide criteria with which to detect such transitions.

## 1. Introduction

This paper analyzes bifurcations of solutions to constrained optimization problems of the form
(1)$$\begin{array}{c} \max_{q\in \Delta }F\left(q,\beta \right)=\max_{q\in \Delta }\left(\sum_{i=1}^{N}f\left(q^{i},\beta \right)\right) \end{array}$$
as a function of a scalar parameter β and a quantizer or classifier $q=(q^{1},\cdots ,q^{N})$ with $q^{i}\in {ℜ}^{K}$. The real-valued function *f* is sufficiently smooth, and Δ is the constraint space of valid quantizers, a convex set of discrete probabilities (simplices).

This type of problem arises in Rate Distortion Theory [1,2], Deterministic Annealing [3] and biclustering [4]. The specific motivations for the abstract problem formulation given in (Equation 1) are the Information Bottleneck [5] and Information Distortion [6] functions
(2)$$\begin{array}{c} \max_{q\in \Delta }F\left(q,\beta \right)=\max_{q\in \Delta }\left(D(q)-\beta I(Y;T)\right). \end{array}$$

These were proposed in [5,7] to analyze the Markov chain X→Y→T in which X→Y, characterized by a probability p(X,Y), is the original system of interest, characterized by its mutual information I(X;Y), and *T* is a simplification (quantized version of) *Y*. Here we work mainly with discrete versions of *Y* and *T*, with cardinalities |Y|=K and |T|=N. Typically N<<K. I(Y;T) is the mutual information between the *K* objects in *Y* and the *N* clusters in *T*. The goal is to cluster *K* objects in *Y* into *N* clusters in *T* given inputs *X* such that the function *F* is maximized in ${\left[q^{i}\right]}_{j}$; the probability that the *j*th element of *Y* is classified as being a member of the cluster with label i∈T. We call such a set of conditional probabilities a *stochastic quantizer*, or just a quantizer, to relate to the vector quantization literature [8]. The annealing parameter β∈[0,∞).

It has been shown that finding hard-clustering solutions to (Equation 2) is NP-complete (combinatorial search) when D(q) is the mutual information I(X;T) [9], as in the Information Bottleneck [5,10,11] and the Information Distortion [7,12,13] methods. Information Bottleneck (IB) approaches are gaining in penetration into multiple scientific and engineering domains [14,15,16,17,18]. As they typically involve the nonlinear optimization problem (Equation 2), there is need for optimization methods for such problems that can avoid the rise in complexity implied by the NP-complete hard-clustering solutions [9]. Originally, Tishby et al. [5] approached this problem with an algorithm inspired by the Blahut–Arimoto approach to solving Rate-Distortion types of problems ([2], Chapter 10). The “self-consistent” equations in [5] optimize both the quantizer and the “relevance” distribution p(x|t). However, unlike the classic Blahut–Arimoto algorithm, which can guarantee convergence to a unique solution to its iterative scheme because of the convex geometry of the two state spaces, the “self-consistent” equations have no such guarantee due to the more-complicated geometry of *three* convex sets over which the optimization is performed, as also noted in [5]. Accordingly, in this work, we use the original optimization problem (Equation 2) over a single variable: the quantizer (conditional probability) q(t|y). It may be possible that a related Blahut–Arimoto style optimization coupled to the bifurcation structure of its gradient flow discussed here can lead to additional insights into this problem, but we consider this beyond the scope of this particular manuscript.

We have investigated the structure of soft-clustering annealing-type methods that reach the hard-clustering solution in the limit of the annealing parameter [19,20] through a series of bifurcations. A bifurcation in this context is a point that is a solution $\left(q^{*},\beta^{*}\right)$ to (Equation 2) such that the number of solutions to (Equation 2) changes in a small neighborhood of $\left(q^{*},\beta^{*}\right)$. Because a bifurcation corresponds to a point at which some of the objects *Y* have just been classified, in the IB literature, a bifurcation is usually referred to as a phase transition. One of the goals of this and related work is to understand why annealing-type algorithms, such as the original optimization heuristics in [5,10], work as well as they do. This can help with designing further optimization heuristics and can assess how close those can get to the global solutions to IB problems. We believe that this amalgamation of optimization theory and dynamical systems theory, as stated in [19,20], can provide a solid foundation with which to address such optimization challenges.

Because of the form (Equation 1) of *F*, it possesses certain symmetries. That is, the value of F(q,β) does not change (is invariant) under arbitrary permutations of the vectors $q^{i}$. In other words, *F* is $S_{N}$-invariant. The form (Equation 1) further implies that the Hessian $d_{q}^{2}F\left(q\right)$ is block diagonal with blocks ${\left\{d_{q^{i}}^{2}f\left(q^{i}\right)\right\}}_{i=1}^{N}$. These conditions are met by the Information Distortion function [6],
(3)$$\begin{array}{c} F_{H}\left(q,\beta \right)=H\left(T|Y\right)+\beta I\left(X,T\right), \end{array}$$
where H(T|Y) is the entropy, and by the cost function used in the original IB method [5],
(4)$$\begin{array}{c} F_{IB}\left(q,\beta \right)=-I\left(Y,T\right)+\beta I\left(X,T\right), \end{array}$$
which is the focus of this manuscript. Both the Information Distortion and Informaton Bottleneck problems have the form given in (Equation 1) and (Equation 2). Importantly, $d^{2}F_{IB}\left(q\right)$ has a “perpetual kernel“ since each block $d^{2}f\left(q^{i}\right)$ has the eigenpair (0, $q^{i}$) for every *q* [20]. In other words, the Hessian $d^{2}F$ is singular for every *q* and every value of β. This makes bifurcation detection challenging because bifurcations can usually be detected by identifying isolated singularities of $d^{2}F$. This degeneracy is a consequence of the translational symmetry of $F_{IB}$: if $k\in \ker d_{q}^{2}F_{IB}\left(q^{*}\right)$, then $F_{IB}\left(q^{*}\right)=F_{IB}\left(q^{*}+tk\right)$ for all t∈ℜ such that $q^{*}+tk\in \Delta$. At bifurcations of solutions to (Equation 4), the translational symmetry never breaks.

To better understand bifurcations of solutions to problems of the form (Equation 1), which includes the problems (Equation 3) and (Equation 4), we consider the gradient flow
$$\left(\begin{array}{c} \dot{q}\\ \dot{\lambda } \end{array}\right)=\nabla L\left(q,\lambda ,\beta \right)$$

Equilibria of this flow correspond to critical points of (Equation 1), where L is the Lagrangian with respect to the constraints imposed by Δ, and λ is the vector of Lagrange multipliers.

Previous work showed that when $d^{2}F$ is generically non-singular, as occurs for the Information Distortion (Equation 3), then there are isolated singularities of $d^{2}L$ that indicate possible bifurcations of solutions to (Equation 1). In this case, an M>1-dimensional $\ker d^{2}F$ necessitates an M−1-dimensional $\ker d^{2}L$, which admits a bifurcation of solutions to (Equation 1) where symmetry breaks from $S_{M}$ to $S_{m}\times S_{n}$ for every m,n>0 such that m+n=M [21].

Here we allow $d^{2}F$ and $d^{2}L$ to be singular for every q∈Δ, as occurs for the Information Bottleneck (Equation 4). That is, the perpetual kernel for $d^{2}F$ implies that $d^{2}L$ also has a perpetual kernel $\ker d^{2}L=K_{p}\left(q\right)$, which means that the eigenvalue crossing condition that must occur at a bifurcation (i.e., $d^{2}L$ must have a zero eigenvalue at a bifurcation) [20] is never satisfied in $K_{p}$. There are a few challenges due to the existence of the perpetual kernel (i.e., degeneracy) of the Information Bottleneck that we address in this paper. First, detecting bifurcations may be problematic because one cannot simply monitor the determinant of either $d^{2}F$ or $d^{2}L$. Second, the standard theory that assures the existence of bifurcating branches, the Equivariance Branching Lemma, cannot be applied directly. Lastly, the spaces that contain the bifurcating solutions are always at least two-dimensional, which makes tracking the bifurcating solutions problematic.

Here we address two of these three challenges. We show that at a bifurcation, new eigenvalue(s) of $d^{2}F_{IB}$ and $d^{2}L$ must cross zero, causing $\ker d^{2}L$ to expand so that $\ker d^{2}L\left(q^{*}\right)=K_{p}\cup K^{*}$, where $K^{*}$ is the span of the eigenvectors with crossing eigenvalues. Instead of detecting bifurcations by the expensive process of monitoring the expansion of $\ker d^{2}L$ (from $K_{p}$ to $K_{p}\cup K^{*}$), we give a simple way to check the eigenvalue crossing condition for annealing problems F=G(q)+βD(q) as in (Equation 2) [20]. We prove the existence of the bifurcating branches by adapting the standard proof for the Equivariant Branching Lemma. This newly developed theory guarantees that bifurcating branches exist in $K^{*}$, are generically pitchforks, and that symmetry breaks from $S_{M}$ to $S_{m}\times S_{n}$. Additionally, we give conditions to check whether the pitchforks are subcritical or supercritical, and how stability of the bifurcating branches relates to optimality in the optimization problem (Equation 1).

## 2. Bifurcation Analysis

### 2.1. Equivariant Branching Lemma

The Equivariant Branching Lemma relates the subgroup structure of a symmetry group Γ with the existence of symmetry-breaking bifurcating branches of equilibria of $\dot{x}=f\left(x,\beta \right)$. Observe that we present a version that does not require absolute irreducibility. For a proof see [22] p. 83.

**Theorem** **1.**(Equivariant Branching Lemma). *Let f be a smooth function f:V×ℜ→V that is* Γ*-equivariant for a compact Lie group* Γ *and a Banach space V. Let* Σ *be an isotropy subgroup of Γ with dimFix(Σ)=1. Suppose that Fix(Γ)={0} and the crossing condition $d_{\beta x}^{2}f\left(0,0\right)x_{0}\neq 0$ for $x_{0}\in \mathrm{Fix}\left(\Sigma \right)$. Then there exists a unique smooth solution branch $\left(tx_{0},\beta \left(t\right)\right)$ to f=0 with isotropy subgroup* Σ*.*

For an arbitrary Γ-equivariant system where bifurcation occurs at $\left(x^{*},\beta^{*}\right)$, the requirement in Theorem 1 that the bifurcation occurs at the origin is accomplished by a translation. Assuring that the Jacobian vanishes, $d_{x}f\left(0,0\right)=0$, can be effected by restricting and projecting the system onto the kernel of the Jacobian. This transform is called the Liapunov–Schmidt reduction (see [23]).

The Equivariant Branching Lemma does not directly apply to yield bifurcating branches for the problem (Equation 1) at *q* for which $d^{2}F$ is singular for the following reasons:$K_{p}$ and $K^{*}$ have independent bases, which implies that each is invariant to the action of $S_{N}$, and so the decomposition $\ker d^{2}L\left(q^{*}\right)=K_{p}\times K^{*}$ shows that $S_{N}$ does not act absolutely irreducibly on $\ker d^{2}F\left(q^{*}\right)$, but it does act absolutely irreducibly on each of these disjoint subspaces separately. This is why we present a version of the Equivariant Branching Lemma that does not require absolute irreducibility.The Liapunov–Schmidt reduction onto $\ker d^{2}L\left(q^{*}\right)$ is clear, but not onto $K^{*}$.$\mathrm{Fix}\left(S_{m}\times S_{n}\right)\cap \ker d^{2}L\left(q^{*}\right)$ is two-dimensional with basis
{(nv,⋯,nv,−mv,⋯,−mv),(ny,⋯,ny,−my,⋯,−my)}, where $v,y\in {ℜ}^{K}$.

We address these issues in the manuscript and show that a small modification of the Equivariant Branching Lemma allows for similar analysis to be successfully applied to Information Bottleneck-style problems such as (Equation 2) with minimal modifications to the original algorithm from [20].

### 2.2. A Gradient Flow

We now lay the groundwork necessary to determine the bifurcations of local solutions to (Equation 1)
$$\max_{q\in \Delta }F\left(q,\beta \right),$$
where $F=\sum_{i=1}^{N}f\left(q^{i},\beta \right)$, which includes as a special case the Information Distortion (Equation 3) and Information Bottleneck (Equation 4) problems. The convex set of discrete conditional probabilities is
$$\Delta :=\left\{q\in {ℜ}^{NK}\;|\;\sum_{i=1}^{N}q_{k}^{i}=1\;\;\forall \;k:1\leq k\leq K\;\mathrm{and}\;q_{k}^{i}\geq 0\;\forall \;i,k\right\}.$$
Due to the form of *F*, it has the following properties:F(q,β) is an $S_{N}$-invariant, real-valued function of *q*, where the action of $S_{N}$ on *q* permutes the component vectors $q^{i}$, i=1,…,N, of q∈Δ.The NK×NK Hessian $d_{q}^{2}F\left(q,\beta \right)$ is block diagonal, where the *i*th K×K block is $d^{2}f\left(q^{i}\right)$.

The Lagrangian of (Equation 1) with respect to the equality constraints from Δ is
(5)$$\begin{array}{c} L\left(q,\lambda ,\beta \right)=F\left(q,\beta \right)+\sum_{k=1}^{K}\lambda_{k}\left(\sum_{i=1}^{N}q_{k}^{i}-1\right). \end{array}$$
The scalar $\lambda_{k}$ is the Lagrange multiplier for the constraint $\sum_{i=1}^{N}q_{k}^{i}-1=0$, and $\lambda \in {ℜ}^{K}$ is the vector of Lagrange multipliers $\lambda ={\left(\lambda_{1},\lambda_{2},\cdots ,\lambda_{K}\right)}^{T}.$ The gradient of the Lagrangian in (Equation 5) is
$$\nabla L:=\nabla_{q,\lambda }L\left(q,\lambda ,\beta \right)=\left(\begin{array}{c} \nabla_{q}L\\ \nabla_{\lambda }L \end{array}\right),$$
where $\nabla_{q}L=\nabla F\left(q,\beta \right)+\Lambda$ and $\Lambda ={\left(\lambda^{T},\;\lambda^{T},\;\cdots \;\;\lambda^{T}\right)}^{T}\in R^{NK}$. The gradient $\nabla_{\lambda }L$ is a vector of *K* constraints
$$\begin{array}{c} \nabla_{\lambda }L=\left(\begin{array}{c} \sum_{i}q_{1}^{i}-1\\ \sum_{i}q_{2}^{i}-1\\ \vdots \\ \sum_{i}q_{K}^{i}-1 \end{array}\right). \end{array}$$
Let *J* be the Jacobian of $d_{q}\nabla_{\lambda }L$
(6)$$\begin{array}{c} J:=d_{q}\nabla_{\lambda }L=\underset{N\;\mathrm{blocks}}{\underset{︸}{\left(\begin{array}{cccc} I_{K} & I_{K} & \cdots & I_{K} \end{array}\right)}}. \end{array}$$
Observe that *J* has full row rank. The Hessian of (Equation 5) with respect to the vector $\left(\begin{array}{c} q\\ \lambda \end{array}\right)\in {ℜ}^{NK+K}$ is
(7)$$\begin{array}{c} d^{2}L\left(q\right):=d^{2}L\left(q,\lambda ,\beta \right)=\left(\begin{array}{cc} d^{2}F\left(q,\beta \right) & J^{T}\\ J & 0 \end{array}\right), \end{array}$$
where 0 is K×K. The NK×NK matrix $d^{2}F\left(q\right):=d_{q}^{2}F\left(q,\beta \right)$ is the block diagonal Hessian of *F* with K×K blocks ${\left\{d^{2}f\left(q^{i},\beta \right)\right\}}_{i=1}^{N}$.

The dynamical system whose equilibria are stationary points of (Equation 1) is the gradient flow of the Lagrangian
(8)$$\begin{array}{c} \left(\begin{array}{c} \dot{q}\\ \dot{\lambda } \end{array}\right)=\nabla L\left(q,\lambda ,\beta \right) \end{array}$$
for L as defined in (Equation 5) and β∈[0,∞). The equilibria of (Equation 8) are points $\left(\begin{array}{c} q^{*}\\ \lambda^{*} \end{array}\right)\in R^{NK+K}$ where
$$\nabla L(q^{*},\lambda^{*},\beta )=0.$$
The Jacobian of this system is the Hessian $d^{2}L\left(q,\lambda ,\beta \right)$ from (Equation 7).

**Remark** **1.**
*By the theory of constrained optimization [24], the equilibria $\left(q^{*},\lambda^{*},\beta \right)$ of (Equation 8) where $d^{2}F\left(q^{*},\beta \right)$ is negative definite on kerJ are local solutions of (Equation 1). Conversely, if $\left(q^{*},\beta \right)$ is a local solution of (Equation 1), then there exists a vector of Lagrange multipliers $\lambda^{*}$ so that $\left(q^{*},\lambda^{*},\beta \right)$ is an equilibrium of (Equation 8) (this necessary requirement is called the Karush–Kuhn–Tucker conditions) such that $d^{2}F\left(q^{*},\beta \right)$ is non-positive definite on kerJ.*


### 2.3. Equilibria with Symmetry

Next, we categorize the equilibria of (Equation 8) according to their symmetries, which allows us to determine when to expect symmetry-breaking bifurcations.

Let $q\in \mathrm{Fix}(S_{M})$ for some 1≤M≤N. Then there exists a partition of {1,2,…,N} into the sets U and R, where |U|=M, so that $q^{i}=q^{j}$ if and only if i,j∈U. Clearly, $d^{2}F$ has *M* identical blocks, ${\left\{d^{f}\left(q^{i}\right)\right\}}_{i\in U}$.

To ease the notation, and without loss of generality, we set
U:={1,…,M}andR:={M+1,…,N}.
To distinguish between the blocks of $d^{2}F$, we write
(9)$$\begin{array}{c} B:=d^{2}f\left(q^{i}\right)\;\mathrm{for}\;\leq 1\leq i\leq M\;\mathrm{and}\;R_{i}:=d^{2}f\left(q^{i}\right)\;\mathrm{for}\;M+1\leq i\leq N. \end{array}$$

As mentioned in the introduction, we assume that for each q∈Δ, each block $d^{2}f\left(q^{i}\right)$ always has at least a one-dimensional kernel with basis vector(s) which depend on *q*. Thus, $\dim\ker d^{2}F\geq N$. At an equilibrium of $\left(q^{*},\lambda^{*},\beta^{*}\right)$ of (Equation 8) where $q\in \mathrm{Fix}(S_{M})$, we consider the following three cases:$\dim\ker d^{2}F\left(q^{*}\right)>N+1$;$\dim\ker d^{2}F\left(q^{*}\right)=N+1$;$\dim\ker d^{2}F\left(q^{*}\right)=N$.

We will show that the first case necessitates a symmetry-breaking bifurcation (Theorem 3). In the second case, there is no bifurcation (Corollary 1). Finally, in the third case, we expect a saddle node [21], a symmetry-preserving bifurcation.

We are able to distinguish between the three cases above by considering which blocks of $d^{2}F\left(q^{*}\right)$ have kernels that have more than one dimension. This motivates the following definition.

**Definition** **1.**
*An equilibrium $\left(q^{*},\lambda^{*},\beta^{*}\right)$ of (Equation 8) is M-singular (or, equivalently, $q^{*}$ is M-singular) if:*

*$q\in \mathrm{Fix}(S_{M})$ so that $q^{i}=q^{j}$ for every 1≤i,j≤M.*

*For B, the M block(s) of the Hessian defined in (Equation 9), kerB has dimension 2 with basis vectors $v,y\in {ℜ}^{K}$. v is associated with the crossing eigenvalues, and y is associated with the constant zero eigenvalue of B.*

*The N−M block(s) of the Hessian ${\left\{R_{i}\right\}}_{i\in ℜ}$, defined in (Equation 9), each have a one-dimensional kernel with basis vector $z\left(i\right)\in {ℜ}^{K}$.*

*The vectors v, y and {z(i)} are linearly independent.*

*The matrix*

(10)
$$\begin{array}{c} A:=B\sum_{i=M+1}^{N}R_{i}^{-}+MI_{K} \end{array}$$

*is nonsingular. $R_{i}^{-}$ is the Moore–Penrose inverse of $R_{i}$. When M=N, we define $A:=NI_{K}$.*



We wish to emphasize that we showed in [21] that requirements 2–5 in Definition 1 hold generically.

A straightforward calculation shows that every block of the Hessian $d^{2}F$ of the Information Bottleneck cost function (Equation 2) is singular for every (q,β), and the basis for $\ker d^{2}f\left(q^{i}\right)$ is $y=q^{i}$ for 1≤i≤M and $z\left(i\right)=q^{i}$ for M+1≤i≤N (Lemma 42 in [25]), which assures that these vectors are linearly independent, as in Definition 1.4. At a bifurcation, the kernels of the identical blocks *B* expand by v as in Definition 1.2. Using the notation above, $y=q^{i}$ for each i∈U, and $z\left(i\right)=q^{i}$ for each i∈R.

### 2.4. The Kernel at a Bifurcation

The equilibria of (Equation 8) change their stability with β, and hence change the solutions to (Equation 1). The changes of stability are determined by the kernel of $d^{2}L\left(q^{*}\right)$ at a bifurcation point $q^{*}$. In this section we show that for any $q\in \mathrm{Fix}(S_{M})$ with M>1, $d^{2}L\left(q^{*}\right)$ has a perpetual kernel $K_{p}$ that is at least M−1 dimensional. The zero eigenvalues associated with the eigenvectors in $K_{p}$ remain constant, so that at a bifurcation point $\left(q^{*},\lambda^{*},\beta^{*}\right)$ of (Equation 8) where $q^{*}$ is *M*-singular, new eigenvalues of $d^{2}L$ must cross zero. Thus, the kernel expands, and the bifurcating directions exist in an “expanded” kernel of $d^{2}L\left(q^{*}\right)$, $\ker d^{2}L\left(q^{*}\right)=K^{*}\times K_{p}$.

We determine a basis for $\ker d^{2}L$ at an *M*-singular $q^{*}$ when M>1. If *q* is 1-singular with a trivial isotropy group (i.e., no symmetery), then $d^{2}L\left(q^{*}\right)$ is non-singular—$K_{p}$ disappears. First, we ascertain a basis for $\ker d^{2}F\left(q^{*}\right)$.

Recall that in the preliminaries, when $x\in {ℜ}^{NK}$, we defined $x^{j}\in R^{K}$ to be the *j*th vector component of x. We now define the linearly independent vectors ${\left\{v_{i}\right\}}_{i=1}^{M}$, ${\left\{y_{i}\right\}}_{i=1}^{M}$, and ${\left\{z_{k}\right\}}_{k=M+1}^{N}$ in ${ℜ}^{NK}$ by
(11)$$\begin{array}{ccc} v_{i}^{j}:=\left\{\begin{array}{c} v\;\mathrm{if}\;1\leq i=j\leq M\\ 0\;\mathrm{otherwise} \end{array}\right.,\; & y_{i}^{j}:=\left\{\begin{array}{c} y\;\mathrm{if}\;1\leq i=j\leq M\\ 0\;\mathrm{otherwise} \end{array}\right.,\; & z_{k}^{j}:=\left\{\begin{array}{c} z(i)\;\mathrm{if}\;M+1\leq j=k\leq N\\ 0\;\mathrm{otherwise} \end{array}\right. \end{array}$$
where $0\in {ℜ}^{K}$, and v and y are defined in Definition 1.2. For example, if M=2 and N=3, then $v_{1}:={\left(v^{T},0,0\right)}^{T}$ and $v_{2}:={\left(0,v^{T},0\right)}^{T}$.

Due to the block diagonal form of $d^{2}F\left(q^{*}\right)$, it is easy to see that the N+M vectors defined in (Equation 11) form a basis for $\ker d^{2}F\left(q^{*}\right)$.

Now, let
(12)$$\begin{array}{ccc} V_{i}=\left(\begin{array}{c} v_{i}\\ 0 \end{array}\right)-\left(\begin{array}{c} v_{M}\\ 0 \end{array}\right), & Y_{i}=\left(\begin{array}{c} y_{i}\\ 0 \end{array}\right)-\left(\begin{array}{c} y_{M}\\ 0 \end{array}\right), & Z_{k}=\left(\begin{array}{c} z_{k}\\ 0 \end{array}\right)-\left(\begin{array}{c} z_{N}\\ 0 \end{array}\right) \end{array}$$
for i=1,⋯,M−1 and M+1≤k≤N−1 where $0\in {ℜ}^{K}$. From (Equation 7), it is easy to see that these three sets of vectors are in $\ker d^{2}L\left(q^{*}\right)$. The next theorem shows that ${\left\{V_{i}\right\}}_{i=1}^{M-1}⋃{\left\{Y_{i}\right\}}_{i=1}^{M-1}$ are a basis for $\ker d^{2}L\left(q^{*}\right)$. This natural partition of the basis vectors shows that $\ker d^{2}L\left(q^{*}\right)$ can be written as $\ker d^{2}L\left(q^{*}\right)=K_{p}\times K^{*}$. According to Definition 1, the “perpetual kernel” corresponding to constant zero eigenvalues of $d^{2}L\left(q^{*}\right)$ is generated by
$$K_{p}=<{\left\{Y_{i}\right\}}_{i=1}^{M-1}>.$$
The part of the kernel that arises at a bifurcation corresponding to eigenvalues crossing zero is
$$K^{*}=<{\left\{V_{i}\right\}}_{i=1}^{M-1}>.$$
The vectors $\left\{Z_{k}\right\}$ do not contribute to $\ker d^{2}L\left(q^{*}\right)$.

**Theorem** **2.**
*If $q^{*}$ is M-singular for 1<M≤N, then $\left\{V_{i}\right\}⋃\left\{Y_{i}\right\}$ from (Equation 12) are a basis for $\ker d^{2}L\left(q^{*}\right)$.*


**Proof.** To show that ${\left\{V_{i}\right\}}_{i=1}^{M-1}⋃{\left\{Y_{i}\right\}}_{i=1}^{M-1}$ span $\ker d^{2}L\left(q^{*}\right)$, let $k\in \ker d^{2}L\left(q^{*}\right)$ and decompose it as
(13)$$\begin{array}{c} k=\left(\begin{array}{c} k_{F}\\ k_{J} \end{array}\right) \end{array}$$
where $k_{F}$ is NK×1, and $k_{J}$ is K×1. Hence,
(14)$$\begin{array}{ccc} & d^{2}L\left(q^{*},\lambda^{*},\beta \right)k=\left(\begin{array}{cc} d^{2}F\left(q^{*},\beta^{*}\right) & J^{T}\\ J & 0 \end{array}\right)\left(\begin{array}{c} k_{F}\\ k_{J} \end{array}\right)=0 & \\ & \Rightarrow d^{2}F\left(q^{*},\beta \right)k_{F}=-J^{T}k_{J} & \\ & Jk_{F}=0. \end{array}$$
Now, from (Equation 6) and the fact that $d^{2}F$ is block diagonal, we have
(15)$$\begin{array}{c} \left(\begin{array}{cccc} d^{2}f\left(q^{1}\right) & 0 & \cdots & 0\\ 0 & d^{2}f\left(q^{2}\right) & \cdots & 0\\ \vdots & \vdots & & \vdots \\ 0 & 0 & \cdots & d^{2}f\left(q^{N}\right) \end{array}\right)k_{F}=-\left(\begin{array}{c} k_{J}\\ k_{J}\\ \vdots \\ k_{J} \end{array}\right). \end{array}$$
We set
(16)$$\begin{array}{c} k_{F}:={\left(x_{1}^{T}\;\;x_{2}^{T}\;\;\ldots \;\;x_{N}^{T}\right)}^{T}, \end{array}$$
and using the notation from (Equation 9), then (Equation 15) implies
(17)$$\begin{array}{ccc} Bx_{i} & = & -k_{J}\;\mathrm{for}\;1\leq i\leq M\\ R_{i}x_{i} & = & -k_{J}\;\mathrm{for}\;M+1\leq i\leq N. \end{array}$$
It follows that $x_{i}=R_{i}^{-}Bx_{1}$ for every M+1≤i≤N. By (Equation 14), we have that $\sum_{i=1}^{N}x_{i}=0$, and so
$$\begin{array}{ccc} & \sum_{i=1}^{M}x_{i}+\sum_{i=M+1}^{N}x_{i}=0 & \\ & \Rightarrow \sum_{i=1}^{M}x_{i}+\sum_{i=M+1}^{N}R_{i}^{-}Bx_{1}+=0. & \end{array}$$
By (Equation 17), for every $1\leq i\leq M,\;\;x_{i}$ can be written as $x_{i}=x_{p}+d_{i}v+e_{i}y$, where $x_{p}\in \mathrm{range}\left(B\right)$, $d_{\eta },e_{\eta }\in ℜ$, and v and y are the basis vectors of kerB from Definition 1.2. Thus,
$$\begin{array}{ccc} B\sum_{i=1}^{M}\left(x_{p}+d_{i}v+e_{i}y\right) & + & B\sum_{i=M+1}^{N}R_{i}^{-}B\left(x_{p}+d_{1}v+e_{1}y\right)=0\\ & \Leftrightarrow & \left(B\sum_{i=M+1}^{N}R_{i}^{-}+MI_{K}\right)Bx_{p}=0\\ & \Leftrightarrow & Bx_{p}=0 \end{array}$$
since $A=B\sum_{i=M+1}^{N}R_{i}^{-}+MI_{K}$ is nonsingular. This shows that $x_{p}=0$. Therefore, $x_{i}=d_{i}v+e_{i}y$ for every 1≤i≤M. Now (Equation 17) shows that $k_{J}=0$, and so $x_{i}\in \ker R_{i}$ for M+1≤i≤N, which implies that
$$\begin{array}{c} x_{i}=c_{i}z\left(i\right)\;\mathrm{for}\;M+1\leq i\leq N. \end{array}$$
Hence, $k=\left(\begin{array}{c} k_{F}\\ 0 \end{array}\right)$, where $k_{F}^{i}=\left\{\begin{array}{c} d_{i}v+e_{i}y\;\mathrm{if}\;1\leq i\leq M\\ c_{i}z\left(i\right)\;\mathrm{if}\;M+1\leq i\leq N \end{array}\right.$, from which it follows that
(18)$$\begin{array}{c} Jk_{F}=\sum_{i=1}^{N}x_{i}=\sum_{i=1}^{M}d_{i}v+\sum_{i=1}^{M}e_{i}y+\sum_{i=M+1}^{N}c_{i}z\left(i\right)=0. \end{array}$$
Linear independence (Definition 1.4) implies that $\sum d_{i}=\sum e_{i}=d_{i}=0$. Thus, $k_{F}=\sum_{i=1}^{M-1}d_{i}\left(v_{i}-v_{M}\right)+\sum_{i=1}^{M-1}e_{i}\left(y_{i}-y_{M}\right).$ Therefore, the linearly independent vectors $\left\{V_{i}\right\}=\left\{\left(\begin{array}{c} v_{i}-v_{M}\\ 0 \end{array}\right)\right\}$ and $\left\{Y_{i}\right\}=\left\{\left(\begin{array}{c} y_{i}-y_{M}\\ 0 \end{array}\right)\right\}$ span $\ker d^{2}L\left(q^{*}\right)$. □

**Corollary** **1.**
*If $q^{*}$ is 1-singular and has isotropy group equal to the identity, then $d^{2}L\left(q^{*}\right)$ is nonsingular.*


**Proof.** If *q* is 1-singular, then $d^{2}F\left(q^{*}\right)$ has a single block *B* with a two-dimensional kernel. The other N−1 blocks $\left\{R_{i}\right\}$ are distinct with one-dimensional kernels. By constructing the vectors as in (Equation 11), we see that $\dim\ker d^{2}F\left(q^{*}\right)=N+1$ with basis vectors $v_{1},y_{1},$${\left\{z_{i}\right\}}_{i=2}^{N}$. Now, following the proof of Theorem 2, we take an arbitrary $k\in \ker d^{2}L\left(q^{*},\lambda ,\beta \right),$ and then decompose k as in (Equation 13) and (Equation 16). The proof to Theorem 2 holds for the present case up until, and including (Equation 18). Linear independence now shows that $d_{i}=e_{i}=c_{i}=0$, which implies that k=0. □

**Remark** **2.** 
*The independent bases given for $K_{p}$ and $K^{*}$ in Theorem 2 imply that each is invariant to the action of $S_{N}$, and so the decomposition $\ker d^{2}L\left(q^{*}\right)=K_{p}\times K^{*}$ shows that $S_{N}$ does not act absolutely irreducibly on $\ker d^{2}F\left(q^{*}\right)$. That is, by definition,*

$$d_{x}r\left(0,\beta \right)\neq c\left(\beta \right)I_{2M-2}.$$

*The explicit bases show that $K_{p},K≅\left\{x\in R^{M}:\sum {\left[x\right]}_{i}=0\right\}$, which implies that $S_{M}$ acts absolutely irreducibly on $K_{p}$ and $K^{*}$ [26]. Thus, $K_{p}$ and $K^{*}$ are each $S_{M}$-irreducible.*


### 2.5. Liapunov–Schmidt Reduction

To show the existence of bifurcating branches from a bifurcation point $\left(q^{*},\lambda^{*},\beta^{*}\right)$ of equilibria of (Equation 8), the Equivariant Branching Lemma requires that the bifurcation is translated to (0,0,0) and that the Jacobian vanishes at bifurcation. To accomplish the former, consider
$$\begin{array}{c} F\left(q,\lambda ,\beta \right):=\nabla L\left(q+q^{*},\lambda +\lambda^{*},\beta +\beta^{*}\right). \end{array}$$
To assure that the Jacobian vanishes, we restrict and project F onto $\ker d^{2}L\left(q^{*}\right)$ in a neighborhood of (0,0,0). This is the Liapunov–Schmidt reduction of F [23],
(19)$$\begin{array}{ccc} r & : & R^{M-1}\times R\to R^{M-1}\\ r(x,\beta ) & = & W^{T}\left(I-E\right)F\left(Wx+U\left(Wx,\beta \right),\beta \right) \end{array}$$
where $Wx+U\left(Wx,\beta \right)=\left(\begin{array}{c} q\\ \lambda \end{array}\right)$. The (NK+K)×(NK+K) matrix I−E is the projection matrix onto $\ker F\left(0,0\right)=\ker d^{2}L\left(q^{*}\right)$ with $\ker\left(I-E\right)=\;\mathrm{range}\;\;d^{2}L\left(q^{*}\right)$. *W* is the (NK+K)×(2M−2) matrix whose columns are the basis vectors $\left\{V_{i}\right\}\cup \left\{Y_{i}\right\}$ of $\ker d^{2}L\left(q^{*}\right)$ from (Equation 12) so that Wx is a vector in $\ker d^{2}L\left(q^{*}\right)$. The vector function U(Wx,β) is the component of (q,λ) that is in range $d^{2}L\left(q^{*}\right)$ such that EF(Wx+U(x,β),β)=0, U(0,0)=0, and
(20)$$\begin{array}{c} d_{x}U\left(0,0\right)=0. \end{array}$$
The system defined by the Liapunov–Schmidt reduction, $\dot{x}=r\left(x,\beta \right)$, has a bifurcation of equilibria at (x=0,β=0), which are in 1−1 correspondence with equilibria of (Equation 8). However, the stability of these associated equilibria is not necessarily the same.

It is straightforward to verify the following derivatives ([23] p. 32), which we will require in the sequel. The (2M−2)×(2M−2) Jacobian of (Equation 19) is
(21)$$\begin{array}{c} d_{x}r\left(x,\beta \right)=W^{T}\left(I-E\right)d_{q,\lambda }^{2}L\left(q+q^{*},\lambda +\lambda^{*},\beta +\beta^{*}\right)\left(W+d_{x}U\left(Wx,\beta \right)\right), \end{array}$$
which shows that
(22)$$\begin{array}{c} d_{x}r\left(0,0\right)=0 \end{array}$$
since $\ker\left(I-E\right)=\;\mathrm{range}\;\;d^{2}L\left(q^{*}\right)$.

Our crossing condition at a bifurcation depends on the matrix of derivatives
(23)$$\begin{array}{c} \frac{\partial^{2}r_{i}}{\partial \beta \partial x_{j}}\left(0,0\right)=d_{\beta }d^{2}L\left[w_{i},w_{j}\right]-d^{3}L\left[w_{i},w_{j},L^{-}d_{\beta }\nabla L\right] \end{array}$$
where the derivatives of L are evaluated at $\left(q^{*},\lambda^{*},\beta^{*}\right)$, and $L^{-}$ is the Moore–Penrose-generalized inverse [27] of $d^{2}L\left(q^{*}\right)$. The vectors ${\left\{w_{i}\right\}}_{i=1}^{2M-2}$ are the basis vectors of $\ker d^{2}L\left(q^{*}\right)$ from Theorem 2.

The (2M−2)×(2M−2)×(2M−2) three-dimensional array of second derivatives is
$$\begin{array}{ccc} \frac{\partial^{2}r_{i}}{\partial x_{j}\partial x_{k}}\left(0,0\right) & = & d^{3}L\left(q^{*},\lambda^{*},\beta^{*}\right)\left[w_{i},w_{j},w_{k}\right]. \end{array}$$
In [21], we showed that $\frac{\partial^{2}r_{i}}{\partial x_{j}\partial x_{k}}\left(0,0\right)=0$ whenever i=j=k≤M−1. In the present case, there are more zero entries since now the basis vectors $\left\{w_{i}\right\}$ are of two types: $w_{i}=V_{i}$ for 1≤i≤M−1 (basis vectors of $K^{*}$); or $w_{i}=Y_{i-M+1}$ for M≤i≤2M−2 (basis vectors of $K_{p}$, see (Equation 12)). We now consider the case when i,j≤M−1 and k>M−1. All other cases are dealt with using a similar argument. Substituting in for $w_{i}$ we have
(24)$$\begin{array}{ccc} \frac{\partial^{2}r_{i}}{\partial x_{j}\partial x_{k}}\left(0,0\right) & = & \sum_{\nu ,\delta ,\eta =1}^{N}\sum_{l,m,n=1}^{K}\frac{\partial^{3}F\left(q^{*},\beta^{*}\right)}{\partial q_{l}^{\nu }\partial q_{m}^{\delta }\partial q_{n}^{\eta }}{\left[v_{i}-v_{M}\right]}_{l}^{\nu }{\left[v_{j}-v_{M}\right]}_{m}^{\delta }{\left[y_{k-M+1}-y_{M}\right]}_{n}^{\eta }\\ & = & \sum_{l,m,n=1}^{K}\frac{\partial^{3}f\left(q^{\nu }*,\beta^{*}\right)}{\partial q_{l}^{\nu }\partial q_{m}^{\nu }\partial q_{n}^{\nu }}\left(\delta_{ij(k-M+1)}{\left[v\right]}_{l}{\left[v\right]}_{m}{\left[y\right]}_{n}-{\left[v\right]}_{l}{\left[v\right]}_{m}{\left[y\right]}_{n}\right). \end{array}$$
The vectors v and y are defined in (2). An immediate consequence of this calculation is that $\frac{\partial^{2}r_{i}}{\partial x_{j}\partial x_{k}}\left(0,0\right)=0$ whenever i=j=k−M+1. Thus, similar arguments show that $\frac{\partial^{2}r_{i}}{\partial x_{j}\partial x_{k}}\left(0,0\right)=0$ whenever:
i=j=k;i−M+1=j=k, i=j−M+1=k, i=j=k−M+1;i−M+1=j−M+1=k, i−M+1=j=k−M+1, i=j−M+1=k−M+1.

Further, we get four different “cubes” of identical entries in the 3-D array. They are:
For i,j,k≤M−1, not all equal, the value of the cube is
$$-\sum_{l,m,n=1}^{K}\frac{\partial^{3}f\left(q^{\nu }*,\beta^{*}\right)}{\partial q_{l}^{\nu }\partial q_{m}^{\nu }\partial q_{n}^{\nu }}{\left[v\right]}_{l}{\left[v\right]}_{m}{\left[v\right]}_{n};$$For i,j≤M−1, not both equal, and j>M−1, the value of the cube is
$$-\sum_{l,m,n=1}^{K}\frac{\partial^{3}f\left(q^{\nu }*,\beta^{*}\right)}{\partial q_{l}^{\nu }\partial q_{m}^{\nu }\partial q_{n}^{\nu }}{\left[v\right]}_{l}{\left[v\right]}_{m}{\left[y\right]}_{n};$$For i≤M−1 and j,k>M−1, not both equal, the value of the cube is
$$-\sum_{l,m,n=1}^{K}\frac{\partial^{3}f\left(q^{\nu }*,\beta^{*}\right)}{\partial q_{l}^{\nu }\partial q_{m}^{\nu }\partial q_{n}^{\nu }}{\left[v\right]}_{l}{\left[y\right]}_{m}{\left[y\right]}_{n};$$For i,j,k>M−1, not all equal, the value of the cube is
$$-\sum_{l,m,n=1}^{K}\frac{\partial^{3}f\left(q^{\nu }*,\beta^{*}\right)}{\partial q_{l}^{\nu }\partial q_{m}^{\nu }\partial q_{n}^{\nu }}{\left[y\right]}_{l}{\left[y\right]}_{m}{\left[y\right]}_{n}.$$

The points above will prove useful when proving that $d^{2}r\left(0,0\right)=0$.

The four-dimensional array of third derivatives of *r* is
(25)$$\begin{array}{ccc} \frac{\partial^{3}r_{i}}{\partial x_{j}\partial x_{k}\partial x_{l}}\left(0,0\right)=d^{4}L\left[w_{i},w_{j},w_{k},w_{l}\right] & - & d^{3}L\left[w_{i},w_{j},L^{-}d^{3}L\left[w_{k},w_{l}\right]\right]\\ & - & d^{3}L\left[w_{i},w_{k},L^{-}d^{3}L\left[w_{j},w_{l}\right]\right]\\ & - & d^{3}L\left[w_{i},w_{l},L^{-}d^{3}L\left[w_{j},w_{k}\right]\right] \end{array}$$
where the derivatives of L are evaluated at $\left(q^{*},\lambda^{*},\beta^{*}\right)$, and $L^{-}$ is the Moore–Penrose-generalized inverse [27] of $d^{2}L\left(q^{*}\right)$.

Since $\ker d^{2}L\left(q^{*}\right)$ is not absolutely irreducible, but $K^{*}$ is, one might try to define a Liapunov–Schmidt reduction by restricting and projecting ∇L onto $K^{*}$. One issue with projecting the reduction onto $K^{*}$ is how to define the projection matrix *E* so that
EF=0 and (I−E)F=0 if  and  only  if F=0
holds and $E\;d_{x}r\left(0,0\right)$ is non-singular in range(E) so that the Implicit Function Theorem assures the restriction (q,λ)=Wx+U(Wx,β), where $U\left(Wx\right)\in \;\mathrm{range}\;(d^{2}L\left(q^{*}\right))$, and $Wx\in K^{*}$ instead of $Wx\in \ker d^{2}L\left(q^{*}\right)$ as in (Equation 19) [23]. Simply ignoring the space $K_{p}$ by considering $U\in \;\mathrm{range}\;(d^{2}L\left(q^{*}\right))$ and $Wx\in K^{*}$ amounts to setting $Wx=k^{*}+k_{p}$ and $k_{p}=0$. Since Wx+U is still embedded in the larger ${ℜ}^{NK+K}$, which contains $K_{p}$, then derivatives are affected by the implicit $k_{p}=0$ constraint. This constraint $P_{K_{p}}\left(q,\lambda \right)=k^{*}+U$ is nonlinear (and may not even be tractable) since $K_{p}$ depends on *q*, where $P_{K_{p}}$ is a projection matrix that depends on *q* (see Theorem 7).

### 2.6. Isotropy Subgroups $S_{m}\times S_{n}$ of $S_{N}$

The decomposition $\ker d^{2}L\left(q^{*}\right)=K_{p}\times K^{*}$ shows that $\mathrm{Fix}\left(S_{m}\times S_{n}\right)\cap \ker d^{2}L\left(q^{*}\right)$ is two-dimensional with basis vectors
$$\{{\left(ny^{T},\ldots ,ny^{T},-my^{T},\ldots ,-my^{T}\right)}^{T},{\left(nv^{T},\ldots ,nv^{T},-mv^{T},\ldots ,-mv^{T}\right)}^{T}\}.$$
Restricted to $K^{*}$, these isotropy subgroups $S_{m}\times S_{n}$ of $S_{M}$ have one-dimensional fixed point spaces. This assures that we can use Theorem 1. We have the following Lemma.

**Lemma** **1.**
*Let M=m+n such that M>1 and m,n>0. Let $U_{m}$ be a set of m classes, and let $U_{n}$ be a set of n classes such that $U_{m}\cap U_{n}=\emptyset$ and $U_{m}\cup U_{n}=\left\{1,\cdots ,M\right\}$. Now define ${\hat{u}}_{\left(m,n\right)}\in {ℜ}^{NK}$ such that*

$$\begin{array}{c} {\hat{u}}_{\left(m,n\right)}^{i}=\left\{\begin{array}{cc} nv & if\;i\in U_{m}\\ -mv & if\;i\in U_{n}\\ 0 & otherwise \end{array}\right. \end{array}$$

*where v is defined as in Definition 1.2, and let*

(26)
$$\begin{array}{c} u_{\left(m,n\right)}=\left(\begin{array}{c} {\hat{u}}_{\left(m,n\right)}\\ 0 \end{array}\right) \end{array}$$

*where $0\in R^{K}$. Then the isotropy subgroup of $u_{\left(m,n\right)}$ is $\Sigma_{\left(m,n\right)}\subset \Gamma_{U}$ such that $\Sigma_{\left(m,n\right)}≅S_{m}\times S_{n}$, where $S_{m}$ permutes $u^{i}$ when $i\in U_{m}$, and $S_{n}$ permutes $u^{i}$ when $i\in U_{n}$. The fixed point space of $\;\Sigma_{\left(m,n\right)}$ restricted to $K^{*}\subset d^{2}L\left(q^{*}\right)$ is one dimensional.*


### 2.7. Bifurcating Branches

**Theorem** **3.**
*Let $\left(q^{*},\lambda^{*},\beta^{*}\right)$ be an equilibrium of (Equation 8) such that $q^{*}$ is M-singular for 1<M≤N, and the crossing condition*

dβd2L[u,u]−d3L[u,u,L−dβ∇L]≠0

*is satisfied. Then there exists bifurcating solutions, $\left(\begin{array}{c} q^{*}\\ \lambda^{*}\\ \beta^{*} \end{array}\right)+\left(\begin{array}{c} tu_{\left(m,n\right)}\\ \beta (t) \end{array}\right)$, where $u_{\left(m,n\right)}\in K^{*}$ is defined in (Equation 26), for every pair (m,n) such that M=m+n, each with an isotropy group isomorphic to $S_{m}\times S_{n}$.*


**Proof.** We mimic the proof of the Equivariant Branching Lemma. Let $u:=u_{\left(m,n\right)}\in \mathrm{Fix}\left(S_{m}\times S_{n}\right)\cap K^{*}$ and let *V* be a matrix with columns composed of the M−1 vectors $\left\{V_{i}\right\}$. Thus, there exists $x_{0}\in {ℜ}^{M-1}$ so that $u=Vx_{0}$. Since $r\left(\mathrm{Fix}\left(S_{m}\times S_{n}\right)\cap K^{*}\right)\subseteq \mathrm{Fix}\left(S_{m}\times S_{n}\right)\cap K^{*}$ (for every $\sigma \in S_{m}\times S_{n}$, r(Vx)=r(σVx) ($u\in \mathrm{Fix}(S_{M}\times S_{n}$) that equals σr(Vx) (by equivariance)), then $r\left(tx_{0},\beta \right)=h\left(t,\beta \right)x_{0}$, where *r* is the Liapunov–Schmidt reduction (Equation 19), and *h* is a polynomial in *t*.Since $K^{*}$ is $S_{M}$-irreducible, then $\mathrm{Fix}\left(S_{M}\right)\cap K^{*}=\left\{0\right\}$ (otherwise, σx=x for some $x\in K^{*}$ for every $\sigma \in S_{M}$, which implies that span(x) is an invariant subspace of $K^{*}$). Now [22] p. 75 shows that r(0,β)=0, and so h(0,β)=0, from which it follows that h(t,β)=tk(t,β). Thus,
(27)$$\begin{array}{c} r\left(tx_{0},\beta \right)=tk\left(t,\beta \right)x_{0}. \end{array}$$
Differentiating with respect to *t* yields
(28)$$\begin{array}{c} d_{x}r\left(tx_{0},\beta \right)x_{0}=\left(k\left(t,\beta \right)+td_{t}k\left(t,\beta \right)\right)x_{0}, \end{array}$$
from which it follows that
$$k\left(t,\beta \right)x_{0}=d_{x}r\left(tx_{0},\beta \right)x_{0}-td_{t}k\left(t,\beta \right)x_{0},$$
and so k(0,0)=0. Furthermore, we see that $d_{\beta }k\left(0,0\right)x_{0}=d_{x,\beta }^{2}r\left(0,0\right)x_{0}\neq 0$ by assumption (see (Equation 23)). This shows that $d_{\beta }k\left(0,0\right)$ is a non-zero eigenvalue of $d_{x}r\left(tx_{0},\beta \right)$ with associated eigenvector $x_{0}$. By the Implicit Function Theorem, k(t,β)=0 has a non-zero unique solution for β=β(t). □

### 2.8. The Crossing Condition for Annealing Problemsn

We next determine how to check the crossing condition in Theorem 3 when *F* is an annealing problem, as in (Equation 2)
F(q,β)=H(q)+βD(q).
First, we show that the crossing condition can be checked in terms of the Hessian of the function *D*. Furthermore, when *G* is strictly concave on $\mathrm{span}(\left\{v_{i}\right\})$, then the crossing condition is always satisfied, and every singularity is a bifurcation.

**Theorem** **4.**
*The crossing condition*

$$d_{\beta }d^{2}L\left[u,u\right]-d^{3}L\left[u,u,L^{-}d_{\beta }\nabla L\right]\neq 0$$

*given in Theorem 3 is satisfied for M-singular q for M>1 if $d^{2}D\left(q\right)$ is either positive or negative definite on $\mathrm{span}(\left\{v_{i}\right\})$.*


**Proof.** Let $x_{0}\in {ℜ}^{2M-2}$ so that $u=Wx_{0}\in \mathrm{Fix}\left(S_{m}\times S_{n}\right)\cap K^{*}.$ Multiplying Equation (Equation 21) on the left by $x_{0}^{T}$ and on the right by $x_{0}$ yields
(29)$$\begin{array}{c} x_{0}^{T}d_{x}r\left(0,\beta \right)x_{0}=u^{T}d_{q,\lambda }^{2}L\left(q^{*},\lambda^{*},\beta +\beta^{*}\right)\left(I_{NK+K}+d_{w}U\left(0,\beta \right)\right)u. \end{array}$$
By Theorem 2, an arbitrary $u\in K^{*}$ can be written as $u=\left(\begin{array}{c} \hat{u}\\ 0 \end{array}\right)$, where $\hat{u}\in \mathrm{span}\left(\left\{v_{i}\right\}\right)\subset \ker d^{2}F\left(q^{*},\beta^{*}\right)$. Substituting this into (Equation 29) and observing that $d^{2}F\left(q^{*},\beta +\beta^{*}\right)=d^{2}G\left(q^{*}\right)+\left(\beta +\beta^{*}\right)d^{2}D\left(q^{*}\right)=d^{2}F\left(q^{*},\beta^{*}\right)+\beta d^{2}D\left(q^{*}\right)$ yields
$$\begin{array}{c} x_{0}^{T}d_{x}r\left(0,\beta \right)x_{0}=\beta \left(\begin{array}{cc} {\hat{u}}^{T}d^{2}D\left(q^{*}\right) & 0^{T} \end{array}\right)\left(I_{NK+K}+\partial_{w}U\left(0,\beta \right)\right)\left(\begin{array}{c} \hat{u}\\ 0 \end{array}\right). \end{array}$$
Differentiating with respect to β, evaluating at β=0, and using (Equation 20) yields
(30)$$\begin{array}{c} x_{0}^{T}d_{x,\beta }^{2}r\left(0,0\right)x_{0}={\hat{u}}^{T}d^{2}D\left(q^{*}\right)\hat{u}, \end{array}$$
which must be non-zero since we assume that $d^{2}D\left(q\right)$ is either positive or negative definite on $\mathrm{span}(\left\{v_{i}\right\})$. □

From (Equation 30), we can get an expression for ξ, the eigenvalue of $d_{x,\beta }^{2}r\left(0,0\right)$ with eigenvector $x_{0}$. Substituting $d_{x,\beta }^{2}r\left(0,0\right)x_{0}=\xi x_{0}$ and observing that $x_{0}^{T}x_{0}=x_{0}^{T}W^{T}Wx_{0}={\hat{u}}^{T}\hat{u}$ yields
(31)$$\begin{array}{c} \xi =\frac{{\hat{u}}^{T}d^{2}D\left(q^{*}\right)\hat{u}}{\left|\right|\hat{u}{\left|\right|}^{2}}. \end{array}$$

The requirement that $d^{2}D\left(q\right)$ is either positive or negative definite on $\mathrm{span}(\left\{v_{i}\right\})$ holds when $d^{2}G\left(q^{*}\right)$ is either negative or positive definite, respectively, on $\mathrm{span}(\left\{v_{i}\right\})$.

**Lemma** **2.**
*Let $d^{2}F\left(q^{*},\beta^{*}\neq 0\right)$ be singular where $q^{*}$ is M-singular such that $d^{2}G\left(q^{*}\right)$ is negative (or positive) definite on $\mathrm{span}(\left\{v_{i}\right\})$. Then $d^{2}D\left(q^{*}\right)$ is positive (or negative) definite on $\mathrm{span}(\left\{v_{i}\right\})$.*


**Proof.** If $u\in \mathrm{span}\left(\left\{V_{i}\right\}\right)\subset \ker d^{2}F\left(q^{*}\right)$, then $u^{T}d^{2}G\left(q^{*}\right)u+\beta^{*}u^{T}d^{2}D\left(q^{*}\right)u=0.$ Since $u^{T}d^{2}G\left(q^{*}\right)u<0$, then $u^{T}d^{2}D\left(q^{*}\right)u>0$. □These results are important for the Information Bottleneck problem (Equation 2), where $d^{2}G\left(q\right)=-d^{2}I\left(Y;Z\right)$ is only non-positive definite on $\ker d^{2}F\left(q^{*}\right)$, but is negative definite on $\mathrm{span}(\left\{v_{i}\right\})$. Thus, every singularity of the Information Bottleneck with $\ker d^{2}L\left(q^{*}\right)=K^{*}\times K_{p}$ is a bifurcation point. The space $K_{p}$ does not contain bifurcating branches since the crossing condition is never satisfied there: for $u\in K_{p}$, ${\hat{u}}^{T}d^{2}G\left(q\right)\hat{u}+\beta {\hat{u}}^{T}d^{2}D\left(q\right)\hat{u}=0+0$ (by Lemma 42 in [25]), and so (Theorem 109, [25]) $\xi =\frac{{\hat{u}}^{T}d^{2}D\left(q\right)\hat{u}}{∥\hat{u}∥}=0$.

### 2.9. Bifurcation Type

Suppose that a bifurcation occurs at $\left(q^{*},\lambda^{*},\beta^{*}\right)$, where $q^{*}$ is *M*-singular. This section examines the type of bifurcation from which emanate the branches
$$\begin{array}{c} \left(\left(\begin{array}{c} q^{*}\\ \lambda^{*} \end{array}\right)+tu,\beta^{*}+\beta \left(t\right)\right), \end{array}$$
whose existence is guaranteed by Theorem 3.

As we showed in [21], the derivative $\beta^{\prime }\left(0\right)\neq 0$ indicates a transcritical bifurcation. If $\beta^{\prime }\left(0\right)=0$, then the bifurcation is degenerate, and if $\beta^{\prime \prime }\left(0\right)\neq 0$, then we have a pitchfork-like bifurcation. Further, $t\beta^{\prime }\left(t\right)<0$ for small *t* indicates a subcritical bifurcating branch, and $t\beta^{\prime }\left(t\right)>0$ for small *t* indicates a supercritical bifurcating branch.

Expressions for $\beta^{\prime }\left(0\right)$ and $\beta^{\prime \prime }\left(0\right)$ are derived as follows. Differentiating k(t,β)=0 from (Equation 27) yields
(32)$$\begin{array}{c} d_{t}k\left(t,\beta \left(t\right)\right)+d_{\beta }k\left(t,\beta \left(t\right)\right)\beta^{\prime }\left(t\right)=0, \end{array}$$
so that $\beta^{\prime }\left(t\right)=-\frac{d_{t}k\left(t,\beta \left(t\right)\right)}{d_{\beta }k\left(t,\beta \left(t\right)\right)}.$ Differentiating (Equation 28) with respect to *t* and then evaluating at t=0 shows that
(33)$$\begin{array}{c} \beta^{\prime }\left(0\right)=\frac{-d_{x}^{2}r\left(0,0\right)\left[x_{0},x_{0},x_{0}\right]}{2||x_{0}{\left|\right|}^{2}\xi } \end{array}$$
where $d_{x}^{2}r\left(0,0\right)\left[x_{0},x_{0},x_{0}\right]=\sum_{i,j,k}\frac{\partial^{2}r}{\partial {\left[x\right]}_{i}\partial {\left[x\right]}_{j}\partial {\left[x\right]}_{k}}\left(0,0\right){\left[x_{0}\right]}_{i}{\left[x_{0}\right]}_{j}{\left[x_{0}\right]}_{k}$ (see (Equation 24)). As shown in the proof to Theorem 3, $\xi =d_{\beta }k\left(0,0\right)$ is the non-zero eigenvalue of $d_{x,\beta }^{2}r\left(0,0\right)$ with eigenvector $x_{0}$.

This expression is similar to the one given in [22] p. 90. The numerator can be calculated via (Equation 24). In [21], we showed that $\beta^{\prime }\left(0\right)=0$. We have the same result in the present case.

**Theorem** **5.**
*If $q^{*}$ is M-singular for 1<M≤N, then all of the bifurcating branches guaranteed by Theorem 3 are degenerate, i.e., $\beta^{\prime }\left(0\right)=0$.*


**Proof.** To show that the numerator of (Equation 33) $d_{x}^{2}r\left(0,0\right)=0$, expand $r_{i}$, the *i*th component of *r*, about x=0,
$$\begin{array}{ccc} r_{i}\left(x,\beta \right) & = & r_{i}\left(0,\beta \right)+d_{x}r_{i}{\left(0,\beta \right)}^{T}x+x^{T}d_{x}^{2}r_{i}\left(0,\beta \right)x+O\left(x^{3}\right)\\ & = & d_{x}r_{i}{\left(0,\beta \right)}^{T}x+x^{T}d_{x}^{2}r_{i}\left(0,\beta \right)x+O\left(x^{3}\right), \end{array}$$
and so
$$\begin{array}{ccc} r_{i}\left(x,0\right) & = & x^{T}d_{x}^{2}r_{i}\left(0,0\right)x+O\left(x^{3}\right). \end{array}$$
Applying the equivariance relation Ar(x,0)=r(Ax,0), where *A* is any element of the group isomorphic to $S_{M}$ that acts on *r* in $R^{M-1}$, and equating the quadratic terms yields
$$A\left(\begin{array}{c} x^{T}d_{x}^{2}r_{1}x\\ x^{T}d_{x}^{2}r_{2}x\\ \vdots \\ x^{T}d_{x}^{2}r_{M-1}x \end{array}\right)=\left(\begin{array}{c} x^{T}A^{T}d_{x}^{2}r_{1}Ax\\ x^{T}A^{T}d_{x}^{2}r_{2}Ax\\ \vdots \\ x^{T}A^{T}d_{x}^{2}r_{M-1}Ax \end{array}\right).$$
By (Equation 24), the diagonal $\frac{\partial^{2}r_{i}}{\partial x_{i}\partial x_{i}}\left(0,0\right)=0$ for each *i* as well as for all of the “multi-diagonals”. This shows that $\frac{\partial^{2}r_{i}}{\partial x_{j}\partial x_{k}}\left(0,0\right)=0$ for every i,j,k (see Theorem 124 in [25]). □When $\beta^{\prime }\left(0\right)=0$, we need to compute $\beta^{\prime \prime }\left(0\right)$ to determine whether a branch is subcritical or supercritical. Differentiating (Equation 32) and setting t=0 shows that $\beta^{\prime \prime }\left(0\right)=-\frac{d_{t}^{2}k\left(0,0\right)}{d_{\beta }k\left(0,0\right)}.$ Differentiating (Equation 28) twice and solving for $d_{t}^{2}k\left(0,0\right)$ shows that
(34)$$\begin{array}{c} \beta^{\prime \prime }\left(0\right)=\frac{-d_{x}^{3}r\left(0,0\right)\left[x_{0},x_{0},x_{0},x_{0}\right]}{3||x_{0}{\left|\right|}^{2}\xi } \end{array}$$
where $Wx_{0}=u=u_{\left(m,n\right)}$. Use Equation (Equation 25) to calculate the numerator, and $\xi =d_{\beta }k\left(0,0\right)$ is the non-zero eigenvalue of $d_{x,\beta }^{2}r\left(0,0\right)$ with eigenvector $x_{0}$, for which we give an explicit expression in (Equation 31) when *F* is an annealing problem.If $\beta^{\prime \prime }\left(0\right)\neq 0$, which we expect to be true generically, then Theorem 5 shows that the bifurcation guaranteed by Theorem 3 is pitchfork-like.

### 2.10. Stability and Optimality

The next Theorem relates the stability of equilibria $\left(q^{*},\lambda^{*},\beta \right)$ in the flow (Equation 8) with optimality of $q^{*}$ in Problem (Equation 1). In particular, if a bifurcating branch corresponds to an eigenvalue of $d^{2}L\left(q^{*}\right)$ changing from negative to positive, then the branch consists of stationary points $\left(q^{*},\beta^{*}\right)$ that are not solutions of (Equation 1). Positive eigenvalues of $d^{2}L\left(q^{*}\right)$ do not necessarily show that $q^{*}$ is not a solution of (Equation 1) (see Remark 1). For example, see page 668 of [21]. A proof of this theorem is given in [21].

**Theorem** **6.**
*For each bifurcating branch guaranteed by Theorem 3, u is an eigenvector of $d^{2}L\left(\left(\begin{array}{c} q^{*}\\ \lambda^{*} \end{array}\right)+tu,\beta^{*}+\beta \left(t\right)\right)$ for sufficiently small t. Furthermore, if the corresponding eigenvalue is positive, then the branch consists of unstable stationary points that are not solutions to (Equation 1).*


### 2.11. Structure of the Symmetry Projection

The matrix $P_{R}\left(q^{*}\right)$ that projects $\left(q,\lambda \right)\in {ℜ}^{NK+K}$ onto $\;\mathrm{range}\;\left(d^{2}L\left(q^{*}\right)\right)\times K^{*}$ by annihilating $K_{p}$ is important for numerical computations for equilibria of IB, since we may want to take each equilibrium found by Newton’s method and take out any part in $K_{p}$. $P_{R}$ is written as a function of *q* since its constitutive vectors y (from Definition 1) depend on *q*. The following theorems clarify the structure of this projection.

**Theorem** **7.**
*$P_{R}\left(q\right)=I-P_{K_{p}}\left(q\right)$, where $P_{K_{p}}=\left(\begin{array}{cc} A & 0\\ 0 & 0 \end{array}\right).$$P_{R}$ and $P_{K_{p}}$ are (NK+K)×(NK+K). The matrix A is NK×NK with $N^{2}$ blocks, ${\left\{A_{ij}\right\}}_{i,j=1}^{N}$, of size K×K, defined by*

$$\begin{array}{c} A_{i,j}=\left\{\begin{array}{cc} \left(M-1\right)yy^{T} & if\;1\leq i=j\leq M\\ -yy^{T} & if\;1\leq i\neq j\leq M\\ 0 & otherwise \end{array}\right. \end{array}$$



For example, if M=N=3, then
$$\begin{array}{c} P_{R}=I-\left(\begin{array}{cccc} 2yy^{T} & -yy^{T} & -yy^{T} & 0\\ -yy^{T} & 2yy^{T} & -yy^{T} & 0\\ -yy^{T} & -yy^{T} & 2yy^{T} & 0\\ 0 & 0 & 0 & 0 \end{array}\right)=I-\left(\begin{array}{cccc} \left(N-1\right) & -1 & -1 & 0\\ -1 & \left(N-1\right) & -1 & 0\\ -1 & -1 & \left(N-1\right) & 0\\ 0 & 0 & 0 & 0 \end{array}\right)⊗yy^{T}. \end{array}$$

**Proof.** Theorem 2 gives the basis of $K_{p}$ as ${\left\{Y_{i}\right\}}_{i=1}^{M-1}$. Let *Y* be the (NK+K)×(M−1) matrix whose columns are the vectors $\left\{Y_{i}\right\}$. For example, if M=3 and N=4, then $Y=\left(\begin{array}{cc} y & 0\\ 0 & y\\ -y & -y\\ 0 & 0\\ 0 & 0 \end{array}\right)$. Thus, the matrix that projects onto $K_{p}$ is $P_{K_{p}}=Y{\left(Y^{T}Y\right)}^{-1}Y^{T}$, and the projection matrix onto $\mathrm{range}(d^{2}L\left(q^{*}\right))$ is $P_{R}=I-P_{K_{p}}$. Direct multiplication of $Y{\left(Y^{T}Y\right)}^{-1}Y^{T}$, with an appeal to Lemma 34 in [25] to compute the inverse, shows that $P_{K_{p}}=\frac{1}{Ny^{T}y}\left(\begin{array}{cc} A & 0\\ 0 & 0 \end{array}\right)$. Dropping the constant yields the result. □

For the Information Bottleneck, the matrix $P_{R}$ is easy to calculate, since $y=q^{i}$ for any i∈U. For example, when $q=q_{\frac{1}{N}}$, then $y^{T}y=\frac{K}{N^{2}}$ and $yy^{T}=\frac{1}{N^{2}}1$, and so
$$P_{K_{p}}=\frac{1}{NK}\left(\begin{array}{ccccc} \left(N-1\right) & -1 & \cdots & -1 & 0\\ -1 & \left(N-1\right) & \cdots & -1 & 0\\ \vdots & \vdots & & \vdots & \vdots \\ -1 & -1 & \cdots & \left(N-1\right) & 0\\ 0 & 0 & 0 & 0 & 0 \end{array}\right)⊗1$$
where 1 is a K×K matrix of 1s. Thus,
$$P_{R}=I_{NK+K}-\left(\begin{array}{ccccc} \left(N-1\right) & -1 & \cdots & -1 & 0\\ -1 & \left(N-1\right) & \cdots & -1 & 0\\ \vdots & \vdots & & \vdots & \vdots \\ -1 & -1 & \cdots & \left(N-1\right) & 0\\ 0 & 0 & 0 & 0 & 0 \end{array}\right)⊗1.$$

**Theorem** **8.**
*The symmetry group $S_{M}$ commutes with the matrix $P_{R}$, which projects onto ${ℜ}^{NK+K}\setminus K_{p}$.*


**Proof.** Let $P:=P_{R}$ be the matrix that projects onto $\;\mathrm{range}\;d^{2}L\left(q^{*}\right)\times K^{*}={ℜ}^{NK+K}\setminus K_{p}$. Since ${ℜ}^{NK+K}=\;\mathrm{range}\;d^{2}L\left(q^{*}\right)\times K^{*}\times K_{p}$, then any $x\in {ℜ}^{NK+K}$ can be decomposed in the respective subspaces as $x=r+k^{*}+k_{p}$. Let σ be an arbitrary permutation matrix in $S_{M}$. Then $\sigma P\left(\begin{array}{c} q\\ \lambda \end{array}\right)=\sigma P\left(r+k^{*}+k_{p}\right)=\sigma \left(r+v\right)$. Since $\;\mathrm{range}\;d^{2}L\left(q^{*}\right)$, $K^{*}$ and $K_{p}$ are all $S_{M}$ invariant; then $\sigma \left(r+v\right)\in \;\mathrm{range}\;d^{2}L\left(q^{*}\right)\times K^{*}$ implies that σ(r+v)=Pσ(r+v), and $\sigma z\in K_{p}$ implies that Pσ(r+v)=Pσ(r+v+z). Thus, σPx=Pσx. □

### 2.12. Visualizations of Sample Resultsn

We illustrate these structures numerically. In [7], we introduced the toy “Four-blob” probability distribution p(x,y) shown in Figure 1.

For the Information Distortion problem (Equation 3) [7,12,13] and the synthetic dataset composed of a mixture of four Gaussians (Figure 1), we determined the bifurcation structure of solutions to (Equation 3) by annealing in β and finding the corresponding stationary points to (Equation 1). A typical run of the derived gradient dynamical system tends to follow the main bifurcation branch $S_{K}\to S_{K-1}$ from the fully symmetric uniform quantizer $q_{\frac{1}{N}}$ (N=4 here) to the fully resolved deterministic quantizer (hard clustering) seen at the end in Figure 2. The permutation symmetry is also obvious there—the value of the cost function does not change if the classes along the vertical axis in *T* are permuted/relabeled. The uniform quantizer $q_{\frac{1}{N}}$ (Item 1 in the figure) plays a special role in the formulation (Equation 3), as it is the *unique* solution to the problem for β=0 as the maximum entropy solution of $\max_{q}H\left(T|Y\right)$. Its loss of stability at the first bifurcation for increasing β can hence be determined analytically and the first bifurcation structure characterized completely. Because of the “perpetual kernel” of the cost function in (Equation 4), the uniform quantizer is just one of a continuous set of “uninformative” quantizers for the IB problem (Equation 4): all {q(t|y):q(t|y)=f(t)}, having constant probability of assignment of each *y* to class *t*, but the assignment weight can be different for different classes. Such a structure does not change the value of the cost function in the IB problem (Equation 4) (but does change it for (Equation 3), which hence does not have this degeneracy). We address the degeneracy of the IB optimization by projecting onto the subspace that has the correct symmetry (i.e., just the uniform quantizer $q_{\frac{1}{N}}$ in this case), as outlined in Remark 2.

A more-thorough structure of the bifurcation diagram, using the analysis presented above, is shown in Figure 3.

Similar to the results we presented in [28], the close-up of the bifurcation at β≈1.038706 in Figure 3B shows a subcritical bifurcating branch (a first-order phase transition) that consists of stationary points of Problem (Equation 1). By projecting the Hessian $\Delta_{q}\left(G\left(q^{*}\right)+\beta D\left(q^{*}\right)\right)$ onto each of the kernels referenced in Theorem 6, we determined that the points on this subcritical branch are **not** solutions of (Equation 1), and yet they **are** solutions of (Equation 2).

Furthermore, observe that Figure 3B indicates that a saddle-node bifurcation occurs at β≈1.037479. That this is indeed the case was proved in [21]. In fact, for any problem of the form (Equation 2), these are the only two types of bifurcations to be expected: pitchfork and saddle-node.

## 3. Conclusions and Discussion

The main goal of this contribution was to show that information-based distortion-annealing problems such as (Equation 2) have an interesting mathematical structure. The most interesting aspects of that mathematical structure are driven by the symmetries present in the cost functions—their invariance to actions of the permutation group $S_{N}$, represented as relabeling of the reproduction classes. Such a structure would hold for any biclustering problem [4] that relies on the intrinsic interaction of a pair of variables for unsupervised clustering. The second mathematical structure that we used successfully was bifurcation theory, which allowed us to identify and study the discrete points at which the character of the cost function changed. The combination of those two tools in [20] allowed us to explicitly compute the value of the annealing parameter β at which the initial maximum at the uniform quantizer $q_{\frac{1}{N}}$ of (Equation 1) loses stability. We concluded that for a fixed system C→Y characterized by p(X,Y), this value is the same for both problems, that it does not depend on the number of elements of the reproduction variable *T*, and that it is always greater than 1. We further introduced an eigenvalue problem that links the critical values of β and *q* for bifurcations, or phase transitions, branching off arbitrary intermediate solutions.

Even though the cost functions $F_{IB}$ (Equation 4) and $F_{H}$ (Equation 3) have similar properties, they also differ in some important aspects. We have shown that the function $F_{IB}$ is degenerate since its constitutive functions I(X;Y) and I(X;T) are not strictly convex in *q*. That introduces additional invariances and singularities that are always preserved, which makes phase transitions more difficult to detect (e.g., the ”uninformative quantizers” q(t|y)=f(t) only) and post-transition directions more difficult to determine. In contrast, $F_{H}$ is strictly convex except at points of phase transitions. The theory we developed here allows us to identify bifurcation directions and determine their stability. Despite the presence of a high-dimensional null space at bifurcations, the symmetries restrict the allowed transition dimensions to multiple co-dimension 1 transitions, all related by group transformations. We achieved that here with three main results. Theorem Equation 8 extended the Equivariant Branching Lemma 1 to the Information Bottleneck case with additional translation invariance. Theorem 4 identified specific conditions at which a bifurcation of the gradient flow (Equation 8) occurs. This condition is computable analytically for the initial bifurcation off the uniform quantizer $q_{\frac{1}{N}}$ and with numeric continuation for subsequent bifurcation. Finally, in Section 2.9, we provided checks for the types of bifurcations that occur, giving conditions to detect saddle-node and pitchfork bifurcations and to determine whether pitchforks are supercritical (second-order phase transitions) or subcritical (leading to first-order phase transitions discontinuous in β). The combination of the three results, together with our previous results in [20], completely characterize the local bifurcation structure of Information Bottleneck-type problems with or without the added translation symmetry.

Despite the further development of the bifurcation formalism for IB presented her, there are still open questions that this manuscript did not resolve. In particular, we still cannot confirm or reject the conjecture that the set of $S_{K}$ symmetric soft-clustering branches connected through symmetry-breaking bifurcations leads to the global hard-clustering optima at β→∞ (multiple equivalent solutions connected by the permutation symmetry of the problem). We believe this is partially due to a discrepancy between practical observations and theoretical results. In particular, we and other practitioners [29,30] note that the only observed symmetry-breaking bifurcations during optimization are of the kind $S_{M}\to S_{M-1}$, while the theory allows for arbitrary $S_{M}\to S_{m}\times S_{n}$ bifurcations. The latter are known to happen and be stable in other biological systems and circumstances [26,31]. This suggests a research approach of comparing and contrasting the different systems that possess the same $S_{N}$ symmetry and symmetry-breaking bifurcations to lead to breakthroughs in this application to optimization in the Information Bottleneck problem.

An additional open problem involves the use of continuous variables, already noted in [5] and explored further in [32,33]. This approach, while important for many real-world problems, involves the application of additional mathematical tools, namely Calculus of Variations [34], which further increases the complexity of an otherwise already complex problem. These difficulties are illustrated in a pair of papers [35,36] that use the continuous formulation. They do present some significant results on conditions of learnability, but both papers manage to only get bounds on β under which learnability (optimal solutions beyond the “uninformative” quantizer) can be achieved. This is possibly due to the presence of continuous spectra in covariance operators of continuous quantizers, something that we avoid by focusing on finite spaces. As a consequence, here and in prior work [20], we show specific values for β for the initial bifurcation from the uniform quantizer, which supports nontrivial clustering. We consider formulation with continuous variables beyond the scope of this manuscript, but look forward to the development of additional techniques to incorporate this important case in the bifurcation framework presented here. Regardless of such developments, any practical problem with numeric optimization will involve discretization of the continuous variables, which effectively converts a continuous problem to the discrete state discussed here.

## Figures and Tables

**Figure 1 entropy-24-01231-f001:**
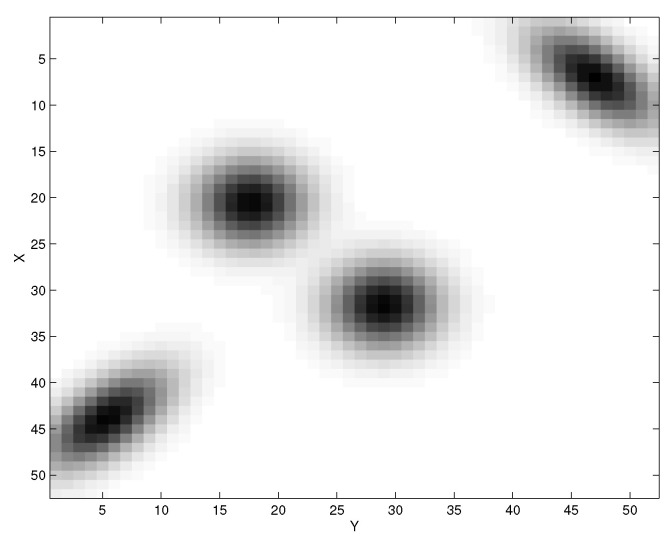
The probability distribution p(x,y) for the “Four-blob” toy problem for a system of interest X→Y. We use this probability to illustrate some results of the bifurcation analysis reported here.

**Figure 2 entropy-24-01231-f002:**
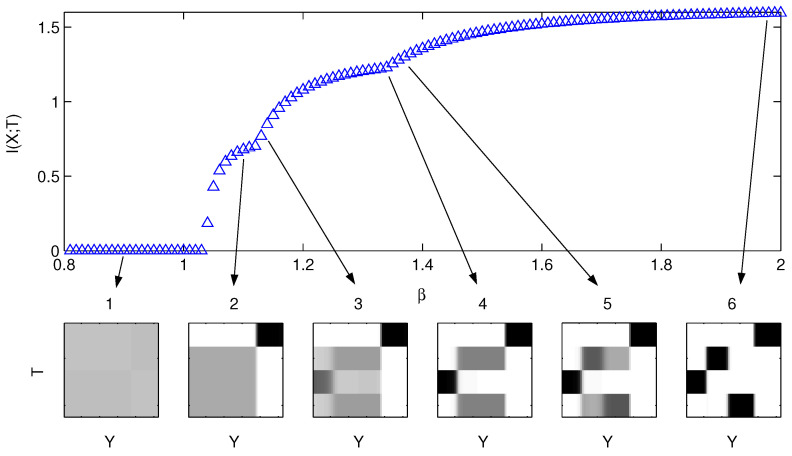
The bifurcations of the solutions $\left(q^{*},\beta \right)$ to the Information Distortion problem (Equation 3). For the mixture of 4 well-separated Gaussians shown in Figure 1, the behavior of D(q)=I(X;T) as a function of β is shown in the top panel, and some of the solutions $q^{*}\left(T|Y\right)$ are shown in the bottom panels. Item 1 shows the uniform quantizer $q_{\frac{1}{N}}$, assigning equal probability of each y∈Y to belong to one of the four clusters in *T*. Subsequent items 2–5 point to a set of partially resolved quantizations, in which subsets of *Y* are assigned with high probability to one (2) or more (3–5) classes (dark colors, close to 1), while other subsets are still unresolved (gray levels), albeit as a higher probability than $q_{\frac{1}{N}}$ (darker gray, as some of the classes are excluded after being resolved for another subset). Item 6 shows an almost fully resolved quantizer at sufficiently high β. They become fully resolved (deterministic; q(t|y)=1 or 0) as β→∞ (not shown).

**Figure 3 entropy-24-01231-f003:**
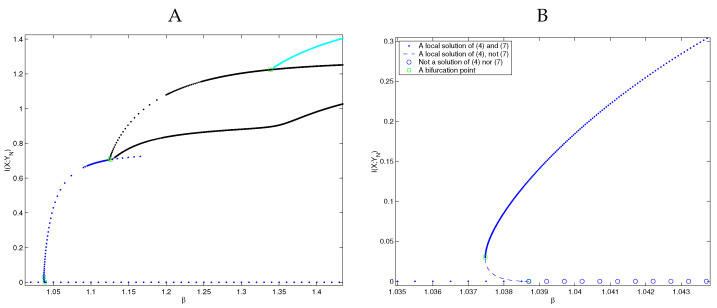
(**A**) The bifurcation structure of stationary points of the Information Distortion problem (Equation 3), a problem of form (Equation 2). We found these points by annealing in β and finding stationary points for Problem (Equation 1) using the algorithm presented in [28]. A square indicates where a bifurcation occurs. (**B**) A close-up of the subcritical bifurcation at β≈1.038706, indicated by a square. Observe the subcritical bifurcating branch, and the subsequent saddle-node bifurcation at β≈1.037479, indicated by another square. We applied Theorem 6 to show that the subcritical bifurcating branch is composed of quantizers that are solutions of (Equation 3) but not of (Equation 1).

## Data Availability

Not applicable.

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
