# Peer review of "Symmetry-Breaking Bifurcations of the Information Bottleneck and Related Problems"

_entropy, 2022, doi:10.3390/e24091231_

Round 1

Reviewer 1 Report

This work provides a characterization of the bifurcations in the solutions of the IB problem for discrete random variables. While I did not verify all of the mathematical proofs in detail, the results appear correct and intuitive. I have a few suggestions which I believe would improve the paper.

- In general, I think the paper has to do better in highlighting how it relates to the Blahut-Arimoto algorithm which is the main algorithm for solving the IB problem.

- While I understand that this work is mathematical, it would be instructive to provide numerical examples to illustrate the results. In particular, the paper seems to suggest tuning the bifurcation from sub to supercritical might be possible. It would be interesting to see if this can happen with synthetic data.

- lines 428: "This condition is computable analytically for the initial bifurcation off the uniform quantizer [...]" The authors should explain why uniform quantizers are special in this context. Is this condition analytically tractable for other trivial encoders, say, all-to-one maps?

lines 429: "[...] with numeric continuation for subsequent bifurcation." A proof-of-principle example of the said numeric continuation would make this claim more convincing.

- The IB method is also applicable to continuous variables. In fact, the structural phase transitions/bifurcations were first observed for continuous variables in Chechik et al., Information bottleneck for Gaussian variables, Journal of Machine Learning Research 6, 165 (2005) [also at NeurIPS (2003)]. I would appreciate a discussion on how the current paper relates to the results for continuous random variables. 

- Since this paper studies the structural change in the IB encoder, I think it would be appropriate to discuss the results in relation to the recent characterizations of IB phase transitions, see, e.g., Wu et al., Learnability for the information bottleneck, Entropy 21, 924 (2019); Ngampruetikorn and Schwab, Perturbation theory for the information bottleneck, NeurIPS (2021).

- The abstract mentions first-order phase transition, yet the paper does not establish whether IB bifurcations are sub or supercritical (apart from giving the mathematical condition). Please clarify.

- lines 263-264 are not readable.

- line 13: typo "we determine the when..."

Author Response

Responses to reviewer notes

We thank the reviewers and editors for helpful suggestions and requests for clarification that only improve this manuscript. Please, see our brief notes below, and more extensive manuscript changes marked up in the submitted file.

2. a paragraph to that effect added in the introduction. (around line 50)
3, 5. we added a new subsection, 2.12, with some visualizations around a toy problem.
4. addressed in the new subsection 2.12, discussion about Figure 2.
6, 14 discussion text added as the last paragraph "Conclusions"

7. Discussion of those two papers added to the Conclusions section. As they also involve continuous state spaces, they are part of the last paragraph response to 6 and 14.

8. Abstract clarified to indicate subcritical bifurcations related to 1-st order transitions, and that we give criteria to detect them. Visualizations now show such a transition for the F_H cost function.
9. revised
10. fixed
13. revised to include more definitions of terms. (revisions marked throughout paper)

15. revised, fixed.

Reviewer 2 Report

This manuscript provides an in depth analysis of the solutions to a class of optimization problems defined in Eq. (1). This class contains special cases that are known and used in statistical / machine learning models, such as the information bottleneck. The aforementioned class displays symmetries with respect to the quantizers  (probability distributions) it optimizes over. The authors build on previous work and investigate the symmetries and the bifurcations they result in, with a particular emphasis on the information bottleneck and information distortion models.

The paper focuses on an important problem that is relevant in a number of areas (algorithm design, statistics, information theory). The results are novel.

The main weakness of the paper is that it relies heavily on previous work and one has to refer to previous publications [18,19] for context and definitions. The current paper is also very condensed which makes it difficult at times to understand where certain claims come from.

For example, it would make the paper more accessible to a wider audience if some definitions were clearly stated (such as “perpetual kernel”, l.56; or “quantizer”, which is, I believe, a discrete probability distribution within the scope of this paper as in [19] and most of IB literature) without making the reader refer to previous work of the authors.

Since the results of the paper concern the discrete case, I was wondering if any connection to the continuous IB can be made (for example the analysis of solutions of the Gaussian IB by Chechik et al.)?

Minor issues:

è Eq. (2) seems to be incompatible with eqs. (3,4) since the former contains  I(Y,T) rather than I(X,T) which is present in the latter (where 3,4 should be both special cases of 2); it is also more frequent to consider T to be a compressed representation of X rather than Y and the IB Markov chain to be T-X-Y rather than T-Y-X, maybe switching X and Y in eq. (2) could simplify all this?

è L.29 “that formally analyses the Markov chain” -> formalizes/introduces the Markov chain?

è L.185--187 spelling: “, when M>1”, exclamation mark?

è L.411 looses -> loses

è L.413 then -> than

è L.415 off -> of

Author Response

(The authors gave the same response as above.)

Round 2

Reviewer 1 Report

The authors have provided a satisfactory response to my questions and concerns. I recommend the publication of this paper in Entropy.